# Plasmid Transfer by Conjugation in Gram-Negative Bacteria: From the Cellular to the Community Level

**DOI:** 10.3390/genes11111239

**Published:** 2020-10-22

**Authors:** Chloé Virolle, Kelly Goldlust, Sarah Djermoun, Sarah Bigot, Christian Lesterlin

**Affiliations:** Microbiologie Moléculaire et Biochimie Structurale (MMSB), Université Lyon 1, CNRS, Inserm, UMR5086, 69007 Lyon, France; chloe.virolle@ibcp.fr (C.V.); kelly.goldlust@ibcp.fr (K.G.); sarah.djermoun@ibcp.fr (S.D.); sarah.bigot@ibcp.fr (S.B.)

**Keywords:** horizontal gene transfer, conjugation in Gram-negative bacteria, phenotypic conversion, drug-resistance dissemination, bacterial biofilms, mobile plasmids, F plasmid

## Abstract

Bacterial conjugation, also referred to as bacterial sex, is a major horizontal gene transfer mechanism through which DNA is transferred from a donor to a recipient bacterium by direct contact. Conjugation is universally conserved among bacteria and occurs in a wide range of environments (soil, plant surfaces, water, sewage, biofilms, and host-associated bacterial communities). Within these habitats, conjugation drives the rapid evolution and adaptation of bacterial strains by mediating the propagation of various metabolic properties, including symbiotic lifestyle, virulence, biofilm formation, resistance to heavy metals, and, most importantly, resistance to antibiotics. These properties make conjugation a fundamentally important process, and it is thus the focus of extensive study. Here, we review the key steps of plasmid transfer by conjugation in Gram-negative bacteria, by following the life cycle of the F factor during its transfer from the donor to the recipient cell. We also discuss our current knowledge of the extent and impact of conjugation within an environmentally and clinically relevant bacterial habitat, bacterial biofilms.

## 1. Introduction

Conjugation was first discovered in 1946 by Edward Tatum and Joshua Lederberg, who showed that bacteria could exchange genetic information through the unidirectional transfer of DNA, mediated by a so-called F (Fertility) factor [1]. It was later realized that the F factor is a replicative extra-chromosomal genetic element, for which they later coined the term plasmid, that can be transferred across the cell membranes of the parental strains. Since this seminal discovery, the identification of a plethora of conjugative elements, including plasmids, conjugative transposons, and integrative conjugative elements (ICEs), has revealed that conjugation is a universally conserved DNA transfer mechanism among Gram-negative and Gram-positive bacteria [2,3]. Conjugation was also shown to be a ubiquitous process that occurs in bacterial communities present in environments such as the soil, on plant surfaces, and in water and sewage, as well as in biofilms and bacterial communities associated with plant or animal hosts [4]. Within these niches, conjugation facilitates the adaptation of bacterial strains by mediating the propagation of advantageous metabolic properties, such as symbiotic lifestyle, virulence, or resistance to heavy metals and antimicrobials. Conjugation is therefore a major driver of the rapid evolution of bacterial genomes [5,6]. This fundamental importance has made conjugation the focus of extensive study over the last decades. Experimental approaches have provided a detailed understanding of the molecular mechanism of conjugational DNA transfer, while systematic sequencing has uncovered the extent of conjugation at the ecological scale.

Conjugative plasmids generally carry all the genes required for their maintenance during the vertical transfer from the mother to the daughter cells, as well as the genes necessary for horizontal transfer during conjugation from the donor to the recipient cell. These functions are encoded by different regions or modules that compose what is generally referred to as the plasmid backbone. Isolation and sequence analysis of an increasing number of conjugative plasmids has revealed considerable diversity in terms of genetic properties and organization. This diversity also indicates that different plasmids might use various regulations, molecular reactions, and strategies to achieve productive conjugational transfer and maintenance.

In this article, we review the key steps of conjugation by following the life cycle of the plasmid during its transfer from the donor to the recipient cell (Figure 1). We focus on the first discovered and extensively described F plasmid, which we use as a paradigm to discuss other conjugative systems in Gram-negative bacteria. The first section describes events occurring within the donor cell, i.e., the expression regulation of the plasmid *tra* genes required for conjugation, the processing of the plasmid by the relaxosome prior to transfer, the composition and function of the conjugative pilus in the mating pair formation process, the central role of type IV coupling proteins (T4CPs), and transfer by the type IV secretion system (T4SS). The second section focuses on the dynamics of the newly acquired plasmid within the recipient cell, i.e., plasmid establishment, which includes protection against host systems dedicated to foreign DNA elimination, early expression of leading genes, and the conversion of the ssDNA plasmid into dsDNA; plasmid maintenance, which includes plasmid replication and segregation; and the eventual phenotypic conversion of the transconjugant into a new donor cell with novel metabolic properties. In the third section, we review our current knowledge of the extent and impact of conjugation within an environmentally and clinically relevant bacterial habitat, bacterial biofilms.

## 2. Within the Donor Cell

### 2.1. Transfer Gene Expression

#### 2.1.1. Regulation of *tra* Gene Expression

The ability of the donor strain to perform conjugation requires the expression of the transfer genes clustered in the *tra* region of the plasmid. The transfer genes encode all the protein factors involved in the elaboration of the conjugative pilus and the T4SS required for the formation of the mating pair, as well as the relaxosome components needed for the processing of the plasmid prior to transfer (Figure 1, step i). The expression of *tra* genes is regulated by several factors, including plasmid and host proteins, cell cycle progression, and environmental conditions. Most *tra* genes are gathered in one operon under the control of the P_Y_ promoter, while *traJ* and *traM* genes are located upstream and controlled by independent promoters (Figure 2) [7]. Transfer gene expression follows a specific regulation cascade that starts with the production of the TraJ protein (Figure 2, Step 1), which activates the P_Y_ promoter and the transcription of the *tra* operon (Figure 2, Step 2). The first gene to be transcribed, *traY*, encodes the TraY regulator protein that activates the P_M_ promoter, resulting in the production of the relaxosome accessory protein TraM (Figure 2, Step 3) [8]. Therefore, this regulation cascade results in the expression of all genes involved in the elaboration of the conjugative pilus, the T4SS, and the relaxosome, which is composed of TraY, TraM, and TraI. It is observed that *tra* genes are normally repressed, presumably to avoid the fitness cost that would be associated with their constitutive expression [9]. It is important to remark that most regulation systems act by modulating the cellular levels or activity of the primary activator TraJ. In most F-like plasmids (R100, R1, R6-5, and ColB2-K77), the expression of *traJ*, and therefore that of other transfer genes, is repressed at the post-transcriptional level by the fertility inhibition system FinOP (Figure 2) [10,11]. FinP is an antisense RNA that is complementary to the stem-loop structures of *traJ* mRNA. FinP binding hides the ribosome binding site and prevents TraJ translation [12,13]. FinO is an RNA chaperon that protects FinP from degradation by RNase E and stabilizes the formation of the FinP–*traJ* mRNA duplex [14,15,16]. Moreover, *tra* gene expression is also regulated by chromosomal-encoded host factors [17]. One such regulation involves the silencing of P_Y_, P_M_, and P_J_ promoters by the chromosome-encoded histone-like nucleoid structuring protein (H-NS) [18,19]. The H-NS copy number per cell varies during growth [20], thus rendering the F plasmid transfer rate growth phase-dependent, i.e., maximum in the exponential phase, reduced in the mid-exponential phase, and mostly abolished in the stationary phase [21,22]. However, during the exponential phase, H-NS repression activity is itself counteracted by the cooperative binding of TraJ and the host protein ArcA (aerobic respiration control of anoxic redox control) to the P_Y_ promoter [23]. In the case of the virulence plasmid pSLT of *Salmonella enterica*, H-NS repression activity also reportedly depends on Dam (DNA adenine methylase) methylation of the DNA [24]. Other examples of host factor-mediated regulation of *tra* gene expression include repression by the RNA binding protein Hfq, which destabilizes both *traJ* and *traM* transcripts [18], and by GroEL chaperone proteins that directly activate proteolysis of plasmid R1 TraJ during the cellular heat shock response [25].

For some other Gram-negative conjugation systems, *tra* gene expression is regulated by quorum-sensing (QS) mechanisms. This is the case for the conjugative tumor-inducing plasmid (pTi), which allows *Agrobacterium* to infect and disseminate within plant hosts. At high cellular density, *Agrobacterium* produces agrocinopine molecules that activate different operons, including the *arc* operon that encodes TraR (unrelated to the F TraR protein), a LuxR-like protein. The binding of TraR to the QS molecule 3-oxo-octanoylhomoserine lactone (OOHL) triggers the transcription of the *trb* and *tra* operons, resulting in the production of the T4SS and relaxosome proteins. The QS lactonase BlcC is also produced, resulting in the degradation of OOHL molecules in the stationary phase or during the carbon and nitrogen starvation associated with host plant death. This regulation provides coordination between the pTi conjugation proficiency and the host state or to the bacterial density within plant tissues during infection [26]. *Pseudomonas aeruginosa* uses QS as a defense mechanism against interspecies conjugation through the production of the QS molecule N-acyl homoserine lactone (AHL), which is involved in the regulation of mechanisms such as virulence, biofilm formation, and metabolism in the *P. aeruginosa* population [27]. AHL produced by *P. aeruginosa* can bind to the *E. coli* LuxR-like transcriptional factor SdiA, which then represses *traI* gene expression and prevents the conjugation of the RP4 broad host range plasmid that is integrated into the chromosome of *E. coli* donors.

Regulating the expression of transfer genes is the chief strategy used to modulate the transfer efficiency of conjugative plasmids. The above examples illustrate that *tra* gene expression is controlled by complex regulatory circuits, which involve the combined activities of plasmids and chromosomal host factors. This tight regulation allows for the control of the transfer efficiency in connection with the plasmid life cycle and the host physiology in response to environmental conditions and populational interactions.

#### 2.1.2. Superspreader Mutations

Over the years, several studies have revealed genetic modifications, so-called superspreader mutations, that dramatically enhanced the conjugation efficiency of conjugative plasmids belonging to diverse incompatibility groups. The first superspreader mutation was characterized in the F plasmid, which carries an IS*3* insertion sequence into the *finO* gene. FinO inactivation destabilizes the FinP-*traJ* mRNA duplex, thus resulting in the upregulation of *traJ* and the constitutive expression of *tra* genes [28]. This naturally occurring mutation accounts for the enhanced transfer efficiency of the F plasmid compared with the related IncF plasmids R100, R6-5, and R1, in which the FinOP regulatory system is still active [29]. More recently, genetically induced superspreader mutations of several resistance plasmids have been isolated in laboratory settings. In the IncI plasmid pESBL, which is associated with extended-spectrum β-lactamase production in *Enterobacteria*, inactivation of the Hft locus triggered the overexpression of conjugative pili and 20-fold enhancement of the transfer efficiency [30,31]. In the *Citrobacter freundii* IncM group plasmid pCTX-M3 that carries the *bla*_CTX-M-3_ gene, the deletion of two genes (*orf35* and *orf36*) resulted in the enhanced expression of *tra* genes and increased plasmid transfer [32]. Another example was reported in the Gram-positive broad host range (Inc18) plasmid pIP501, which is involved in the propagation of vancomycin resistance from *Enterococci* to methicillin-resistant strains of *Staphylococcus aureus.* In this case, the deletion of the *traN* gene encoding the small cytosolic protein TraN (unrelated to the F TraN protein) resulted in the upregulation of transfer factors and the enhancement of the transfer efficiency [33].

Inducing the overexpression of plasmid transfer genes might not be the only way through which *superspreader* mutations increase the transfer efficiency of conjugative plasmids. It was shown that insertion of the Tn*1999* transposon into the *tir* (transfer inhibition of RP4) gene of the IncL/M-type plasmid pOXA-48a, responsible for the dissemination of specific extended-spectrum β-lactamase genes in *Enterobacteriaceae*, increases the transfer efficiency by 50–100-fold without affecting *traM* expression levels [34]. The mechanism by which the inactivation of the Tir protein enhances transfer efficiency remains to be elucidated. These studies show that superspreader mutations can emerge by various mechanisms in different conjugative plasmids and have the potential to aggravate the spread of drug resistance plasmids among bacterial organisms.

### 2.2. Conjugative Pilus, Mating Pair Formation, and Stabilization

#### 2.2.1. F-Pilus Structure and Biosynthesis

Bacterial conjugation is a contact-dependent horizontal gene transfer mechanism that involves a conjugative pilus associated with a T4SS. Electron microscopy imaging was instrumental in analyzing the morphology of numerous conjugation pili encoded by plasmids belonging to different incompatibility groups [35,36,37,38]. These studies revealed that pili fall into two main morphological categories—thin flexible and thick rigid, which influence the ability to support conjugation in liquid or on a solid surface. F encodes a thin flexible pilus, which has a tubular structure ~8 µm in diameter and up to ~20 µm in length and which is constituted by a helicoidal arrangement of a unique protein subunit, the F pilin or TraA [39,40,41,42,43,44] (Figure 1, step i). The *traA* gene encodes a 121 amino acid pro-pilin peptide that subsequently processed into a 70 amino acid F Pilin [45,46]. The maturation process involves TraQ and TraX proteins [47]. The TraQ chaperone-like protein binds transiently to the TraA pro-pilin precursor, thus allowing its accumulation in the inner membrane by an ATP-dependent pathway [48] and imparting it with the right conformation for a signal peptide cleavage. The processing of pro-pilin into mature pilin requires both cleavage by the host leader peptidase B (LepB) and N-terminal acetylation by TraX [49,50]. This maturation process ensures the availability of pilin subunits in the inner membrane before the assembly of the pilus by TraE, TraK, TraB, TraV, TraC, TraW, TraG, TraF, TraH, TraL, and TrbC encoded by the transfer region [51,52,53]. Mutational experiments have shown that this set of proteins can separate into different functions. Briefly, TraE, K, C, G, and L are responsible for the assembly of the tip, while TraB, V, W, F, and H are important for the pilus extension, and TrbI is required for pilus retraction [52,53]. The pathway of F-pilus biosynthesis has been extensively reviewed [52,54] and is not detailed here (Table 1).

#### 2.2.2. Pilus Biological Function

The role of the F-pilus in conjugation has been actively debated. It was first proposed that the F-pilus extends to contact the recipient cell and then retracts to bring together the donor and recipient cells and form the mating pair [41,67,68]. This idea was convincingly supported by the direct visualization of the F-pilus dynamics in live cells, using a fluorescently labeled R17 bacteriophage that specifically binds along the pilus sides [69]. This work showed that donor cells produce flexible pili that continuously undergo cycles of extension and retracting, thereby probing the surroundings, regardless of the absence or presence of recipient cells. However, when contact is established with a recipient, pilus retraction draws the cells together, resulting in the formation of a mating pair [69]. In liquid culture, the pili mediate the formation of larger mating aggregates that contain donor and recipient cells in tight wall-to-wall contact [70,71].

Allowing the formation of wall-to-wall contact between mating partners might not be the only role of the F-pilus, which was also proposed to serve as a channel through which single-stranded DNA is transferred during conjugation between distant donor and recipient cells [72]. Undeniably, the pilus axial hole has a diameter (30 Å) that is large enough to accommodate the DNA molecule [41,44]. However, only a few reports provide evidence for conjugational DNA transfer between mating partners that are spatially separated from each other. It was shown that DNA transfer could occur between a donor and recipient that are separated by a 6 micron membrane with pores 0.01–0.1 micron in diameter [73]. Furthermore, microscopy imaging provided some evidence that DNA can be acquired by recipient cells that are not in direct contact with a donor cell [74]. Nevertheless, the F-pilus’ ability to transport DNA is still questioned and awaits the clear visualization of DNA transfer between distant donor and recipient cells that are only connected by a pilus.

#### 2.2.3. Factors Involved in the Specificity of Donor–Recipient Interactions

The ability of the pilus to establish contact between donor and recipient cells can be considered the first rate-limiting step in the conjugation process and a key determinant of plasmid host range specificity. In the 1970s, numerous studies attempted to identify the specific recipient receptor required for F plasmid transfer [70,75,76,77,78,79,80]. The results revealed that mutations localized in the fourth external loop of the major porin OmpA or those that alter the inner core composition of the lipopolysaccharide (LPS) affect the transfer of the F plasmid and other IncF-like plasmids, such as R386, R538-1drd, and R1-19, but not the IncFII-type plasmid R100-1 [77,80,81,82]. Analysis of several *ompA* and LPS mutants revealed that they do not affect pili binding but result in defective mating pair stabilization [83]. Further investigation excluded that TraA is the donor component responsible for specific recognition of the recipient receptors [84] and uncovered the mating pair stabilization function of the outer membrane protein TraN, whose three external loops have been reported to interact with OmpA and LPS [85,86,87]. These findings indicate that *ompA* and LPS mutations do not alter the conjugation efficiency of the closely related IncFII-type plasmid R100, in which the amino acid sequence of the TraN central region is highly divergent from that of F-encoded TraN [59,77,80]. Mating pair stabilization additionally involves the multifunctional inner membrane protein TraG, of which the N-terminal part also plays a role in piliation and surface exclusion [59,88,89,90].

OmpA or LPS receptor factors cannot be considered strictly essential to F conjugation since their mutation only decreases transfer efficiency by 2–3 log. Moreover, mating pair stabilization defects can be overridden by performing mating on solid media, suggesting that interactions with OmpA or LPS are needed to stabilize the mating pair formation only in liquid mating [77,81,83]. One might then ask, is the F plasmid an exception to the need for recipient factors in some conditions? A recent study in *Klebsiella pneumoniae* identified the outer membrane OmpK36 homolog of the *E. coli* outer membrane OmpC as a receptor that mediates conjugation of the IncFII plasmid pKpQIL [91]. As observed in the case of the F plasmid, TraN_pKpQIL_ determines the OmpK36-receptor specificity, while complementing a *traN**_pKpQIL_* mutant with TraN_R100_ abolished this dependence, demonstrating that recipient receptors might be highly specific to the transferred plasmid. In addition to the IncF-type, the conjugation process of IncI-type plasmids was also shown to be sensitive to LPS mutations, and interestingly, some LPS mutations that affect IncI plasmid transfer do not impact the entry of the F plasmid, while others affect both types of plasmid [76,92]. Recently, the PilV adhesin encoded by the IncI1-type plasmid R64 was identified as the donor factor that binds to LPS in the recipient cell [93]. This adhesin is thought to localize to the tip of the thin type IV pilus that is required only in liquid mating conditions, rendering the interaction between PilV and LPS important only under these conditions, as observed for TraN and OmpA or LPS interactions in the case of the F plasmid. In contrast, no such recipient receptors could be identified for the broad host range plasmids RP4 and R388. Indeed, the *ompA* and LPS recipient mutants, which drastically decrease the efficiency of F conjugation, do not affect RP4 conjugation efficiency [94]. Large-scale mutant screening using the *E. coli* Keio collection or random *E. coli* insertion mutant library failed to identify recipient mutants that affect the transfer of the plasmids RP4 [95] or R388 [96].

Remarkably, some broad host range IncP-like plasmids can also be transferred in archaea [97] and eukaryotes such as yeast [98] and mammalian cells [99]. Although the efficiency of conjugation varies among types of recipient cells, these findings strongly suggest that plasmid transfer does not require any specific factors or active mechanisms on the recipient side. Alternatively, a “shoot and pump” conjugation model envisages that the type IV secretion apparatus could act like a syringe that is able to inject DNA into any kind of membrane, using the pilus as a needle [100]. Perforation of the cell wall bilayer of the recipient could be achieved by force or by dedicated enzymatic activity exposed at the pilus tip. The lack of a requirement for specific receptors on the surface of the target cell is not an exception to the conjugative T4SS. Indeed, although the structural components of the type VI secretion system machinery have been widely documented, no studies have yet characterized genetic factors that can act as target receptors on the prey cell surface.

### 2.3. Plasmid Processing by the Relaxosome

The initiation of conjugation requires the assembly and activity of a protein complex, the relaxosome, that allows the processing of the plasmid before DNA transfer (Figure 1, step ii). Plasmid processing involves a site- and strand-specific DNA cut (nick) at the *nic* site located in the origin of transfer (*oriT*) and the extrusion of the single-stranded DNA that will be transferred (T-strand) [101,102,103]. In the F system, these two reactions are performed by the multifunctional TraI relaxase protein, which has both a transesterase domain that catalyzes the nic reaction and a DNA helicase domain that unwinds the plasmid DNA [64,104,105,106,107,108,109]. Crucially, TraI recruitment and activity are governed by auxiliary proteins, including the integration host factor (IHF) and the plasmid-encoded TraY and TraM proteins, which have distinct roles in the relaxosome formation and activity at oriT [110,111,112]. The binding of IHF and TraY to their respective cognate binding sites *ihfA*/*ihfB* and *sbyA* located in oriT modulates the architecture of the DNA, thereby stimulating the loading of TraI [113,114,115,116,117]. The TraM protein regulates its own expression by binding to the *sbmA* and *sbmB* sites, located in the PM promoter, and stimulates the DNA relaxation reaction through direct interaction with TraI after binding to the *sbmC* site located in the *oriT* region [118,119,120,121,122]. The TraI nicking reaction involves a catalytically active Tyr residue [65,123] and results in the relaxation of the plasmid dsDNA, where the 5′-phosphate (P) end of the nicked strand (or T-strand) remains covalently bound to TraI (Figure 1, step iii) [106,108,124,125,126,127]. After the nicking reaction, the circular ssDNA conjugative plasmid is converted into dsDNA by Rolling Circle Replication (RCR) in the donor, while the linearized T-strand DNA bound to TraI at the 5′ end is transferred into the recipient cell through the conjugative pore (Figure 1, step iv).

### 2.4. Initiation of Rolling Circle Replication in the Donor Cell

The rolling circle replication (RCR) mechanism is employed for the vegetative replication of some bacterial plasmids and has been very well reviewed [128,129,130]. RCR is key to the transfer process of many Gram-negative and Gram-positive conjugative plasmids but also to the infection cycle of other mobile genetic elements, such as DNA/RNA viruses and bacteriophages [123,131]. The RCR reactions involved in vegetative replication or in plasmid transfer are very similar. The initiation and termination of RCR reactions, performed by the Rep protein during vegetative plasmid replication, are achieved by the relaxase protein during conjugation. Indeed, Rep and relaxases serve closely related functions, primarily allowing RCR initiation by nicking the double-strand DNA at the *dso* site (double-stranded origin) or the *oriT* site, respectively [132]. The nicking reaction generates a 5′-P end that remains covalently bound to Rep or TraI and a 3′OH end used as a primer for the host DNA polymerase III. While DNA polymerase III performs leading strand elongation, the parental double helix is unwound, and RCR ends with a second nicking reaction that releases the newly synthesized DNA strand (Figure 1, step v). In the case of vegetative RCR, DNA unwinding is performed by a host DNA helicase recruited by the Rep protein, while Rep itself ensures termination and the second nicking reaction. One major aspect of conjugation-associated RCR is that replication of the two ssDNA strands occurs in different cells, i.e., the leading strand is replicated in the donor, while the T-strand (lagging strand) is transferred and replicated in the recipient cell (Figure 1). Because the relaxase that initiates the nicking reaction is transferred together with the T-strand [65,133,134], a second relaxase protein is required in the donor to perform DNA unwinding as well as the second nicking reaction [3,65,123]. Consistently, biochemical assays show that two relaxase molecules bind to *oriT*: one associated with the 5′ end that is in an open transesterase conformation and one associated with the 3′ end that is in a closed helicase conformation [135].

### 2.5. T4CP Connects the Relaxosome to the T4SS

After processing by the relaxosome complex, the nucleoprotein complex, composed of the T-strand and the covalently bound TraI, needs to be recruited to the conjugative pore for transfer (Figure 1, step iii). This connection is mediated by the interaction between the relaxosome and the Type IV Coupling Protein (T4CP) located at the cell membrane [3,100]. All conjugative systems have their own T4CPs, such as TraD, TraG, and TrwB for the F, RP4, and R388 plasmids, respectively. T4CPs are not required for pilus production or DNA processing, yet they are key to substrate specificity [136]. Our understanding of the molecular interactions required for specific substrate recognition and translocation is still incomplete. A great deal of information has been provided by comparing the structure of F-like T4SS [137] to various T4SSs involved in protein or nucleoprotein transport [138,139,140,141]. It appears likely that conjugation systems are derived from ancestral protein translocation machinery that evolved to coincidently translocate DNA. In this view, the T4CP would serve as the substrate receptor that interacts with one or several relaxosome components to recruit the T-strand to the T4SS. In the case of the F plasmid and some other plasmid systems, it is well established that TraD interacts with the TraM relaxosome protein [142,143,144,145,146]. Interaction between the T4CP and the relaxase has been demonstrated for RP4 [147], R1 [1] and R388 plasmids [148]. Such interaction has been speculated in the F system but remains elusive [107,149].

T4CPs are DNA-dependent ATPases anchored to the cell membrane via their N-terminal domain and have been shown to interact with the T4SS components in R27 [150] and R388 plasmid systems [148]. T4CPs show similarities to membrane-anchored ring DNA translocases, such as SpoIIIE and FtsK, which are involved in chromosome DNA translocation during sporulation and cell division, respectively [100,151,152]. T4CP binds non-specifically to DNA, with a higher affinity for ssDNA [147,153,154], on which it forms oligomers with enhanced ATPase activity [155,156].

Altogether, these findings led to a model in which membrane-anchored T4CPs interact directly with the relaxosome and form hexameric structures on the T-strand, which actively translocate through the conjugation pore during transfer. However, it remains unclear whether a signal is required to activate the coupling function of the T4CP in a donor cell in which the T4SS and the relaxosome are already assembled and functional [3]. It has been suggested that the stability of the TraD oligomeric complex depends on an as yet unidentified F-encoded protein, which could then be a key regulator of plasmid transfer activation [155]. It has also been suggested that the formation of the mating pair could transduce a signal to activate the T4CP and trigger the transfer of the processed T-strand [3]. Notably, it was shown that the relaxase has to be unfolded to be translocated into recipient cells [134]. In *A. tumefaciens,* as in many other Gram-negative and Gram-positive systems, the unfolding of translocated proteins is proposed to be performed by the VirB11-like ATPase, which is absent from the F plasmid system [52,140,157]. However, one can reasonably presume that TraI unfolding also requires ATPase activity that involves one of the ATPases of the F T4SS.

## 3. Within the Recipient Cell

### 3.1. Plasmid Circularization by TraI

The relaxase is transferred to the recipient, where it is refolded and able to perform several activities required for the completion of the conjugation process (Figure 1, step a) [65,133,158]. The helicase activity of the internalized relaxase is thought to perform 5′ to 3′ tracking of the T-strand. Pulling by the relaxase from the recipient together with pushing by the T4CP from the donor presumably facilitates the transport of the T-strand through the conjugation pore [3]. Once both extremities of the *oriT* are brought together in the recipient, the relaxase performs the joining reaction, resulting in the recircularization of the ssDNA plasmid (Figure 1, step b) [65,133,158,159]. An alternative model proposes that the nicking of the newly synthesized *oriT* occurs in the donor cell before transfer [160]. There is no evidence for the requirement of additional host or plasmid factors in the circularization of the internalized T-strand. After completion of the recircularization reaction, the recipient cell possesses a single-stranded circular copy of the conjugative plasmid.

### 3.2. Avoiding Host Defense Systems against Foreign DNA

The newly acquired ssDNA conjugative plasmid might be considered as foreign DNA, against which host bacteria have developed defense mechanisms, such as restriction modification, exonucleases, and recombination system or adaptive immunity, such as the CRISPR-Cas system [161]. Despite these defense mechanisms, horizontal gene transfer plays an important role in genomic evolution (5–6% of bacterial genomes and up to 20% in some organisms) [6,162,163], implying that transferrable plasmids have evolved adaptive mechanisms to counteract these host defenses.

The restriction modification system (RM) is a ubiquitous defense mechanism found in 90% of sequenced bacterial genomes and other prokaryotes and is based on restriction enzymes and methylation [164]. Restriction modification mechanisms are described as a “primitive immunity system” against exogenous DNA [165]. These systems are based on restriction enzymes that specifically target unmethylated dsDNA sequences located in the newly acquired mobile genetic elements, while the host DNA is protected by methyl groups added to specific adenine or cytosine residues [165]. Whether the RM system can target the ssDNA plasmid before the complementary strand synthesis remains unclear. However, plasmids have evolved various strategies to counteract enzymatic DNA degradation upon entry into the new host cell. IncN, IncI, and IncF plasmids encode ArdA and ArdB proteins (alleviation of restriction of DNA) that directly inhibit the REase (Restriction Endonuclease) by mimic DNA sequences, thus competing for enzyme targets [166,167,168]. IncW plasmids encode the ArdC protein that protects the transferred T-strand by transiently blocking the restriction sites [169]. More recently, it was shown that an *hde* operon (host defense evasion) of IncI plasmids encodes two genes involved in anti-RM (*vcrx089* and *vcrx090*) [170]. In addition to the production of inhibitory proteins, some plasmids have completely lost these restriction sites, as in the case of the RP4 plasmid [171].

CRISPR-Cas immune systems (Clustered Regularly Interspaced Short Palindromic Repeats and the CRISPR-associated protein) represent another defense mechanism against foreign DNA. CRISPR-Cas systems, found in ~45% of bacterial and up to 84% of Archaea genomes [172,173], have been described as being protective against infection by bacteriophages and, more recently, against plasmid acquisition [174]. It was then discovered that some phages encode an anti-CRISPR Acr protein that inhibits the activity of the CRISPR-Cas system [175,176]. Importantly, anti-CRISPR Acr *loci* have been identified in conjugative elements and plasmids of *Listeria*, *Enterococcus*, *Streptococcus,* and *Staphylococcus* [177]. These *loci* encode the CRISPR-Cas inhibitors AcrIIA16–19, which prevent exogenous DNA nicking mediated by the Cas9 enzyme *in vivo*. Mahendra et al. have also shown that conjugation of a Cas9-targeted plasmid of *E. faecalis* was possible in the presence of these CRISPR-Cas inhibitors. Encoding Acr-like proteins is therefore an efficient strategy by which conjugative plasmids facilitate their dissemination by avoiding degradation by the host CRISPR-Cas immune system. Another strategy is for conjugative plasmids to encode a Bet/Exo system able to repair double-strand breaks caused by CRISPR-Cas during conjugation, as recently reported for the IncC plasmid pVCR94 [170]. Expression of these genes inside recipient bacteria after the acquisition of the *V. cholerae* pCVR94 plasmid enables survival against exogenous DNA defense mechanisms without the involvement of anti-CRISPR-Cas proteins.

### 3.3. Role of the Leading Region Genes and Conversion of the ssDNA Plasmid into dsDNA

#### 3.3.1. Early Expression of the Leading Region Genes

The leading region of conjugative plasmids is the first to be transferred into the recipient cell during conjugation (Figure 1, step a) [57]. The F plasmid leading region is well conserved, with a size of 13 kb and a location that is directly adjacent to the *oriT*, and encodes for at least eight proteins [178], including a homolog to the chromosomal single-strand binding protein SSB (SSB_C_), PsiB, and other proteins of unknown function. Importantly, plasmid *ssb* (*ssb_P_)* and *psiB* genes are expressed early upon entry of the plasmid into the recipient bacteria but not in the donor cells [179]. Similarly, RT-PCR studies showed that the expression of the *psiB* and *ardA* genes of the IncI1 plasmid begins 5 min after transfer initiation [180]. These observations suggested that leading region genes could be expressed rapidly from the newly acquired plasmid in single-stranded form, before its conversion into dsDNA. It was later shown that the leading region contains a specific 328 bp *Frpo* region (for F plasmid RNA polymerase), which, when single-stranded, can form a stem-loop structure presenting −10 and −35 double-stranded boxes that are recognized by the host RNA polymerase, which initiates the synthesis of RNA primers in vitro [181]. It was therefore proposed that *Frpo* may serve as a single-stranded promotor that allows the early expression of the leading region genes (Figure 1, step a).

*Frpo* was also proposed to direct the single-strand to double-strand conversion of the F plasmid. In vitro assays showed that the RNA primers synthesized by the RNA polymerase from *Frpo* persist as an RNA–DNA duplex that is recognized by host DNA polymerase III to initiate complementary strand synthesis [181]. *Frpo*-type sequences, also termed *ssi* for single-strand initiation sequence, are found on various conjugative plasmids, including R6K, R100, ColE1, ColE2, Col1B, and RSF1010 [182,183,184,185], and are functionally comparable to *sso* sequences (single-stranded origin) involved in the rolling circle replication mechanism [123,129,186]. These findings are consistent with the previous observation that complementary strand synthesis of the ssDNA F plasmid inside the recipient bacteria involves a cooperative mechanism between host RNA polymerase and DNA polymerase III [187].

Altogether, these findings led to a model in which *Frpo* can help to initiate early gene expression and the DNA synthesis reaction that converts the ssDNA plasmid into dsDNA duplex immediately upon entry of the T-strand into the recipient cell (Figure 1, step a–c). Whether *Frpo* performs these functions during conjugation in vivo remains to be demonstrated.

#### 3.3.2. PsiB Inhibits the SOS Response

In the recipient cell, the presence of abnormal amounts of ssDNA, usually associated with DNA damage, results in the induction of the SOS response [188,189]. More precisely, the loading of the RecA recombination protein onto ssDNA results in the formation of the presynaptic filament, which stimulates the autocatalytic cleavage of LexA, the repressor of the SOS regulon. The SOS response triggers the induction of the division inhibitor SulA, resulting in cell filamentation and potentially the death of the transconjugant cell. SOS also induces the production of nucleases and other DNA processing factors that could provoke the degradation or mutation of the transferred ssDNA or its processing as a recombination intermediate [190]. To counteract these effects, several conjugative plasmids, including F, encode the PsiB protein (plasmid SOS inhibition), which inhibits SOS induction [191]. The depletion of *psiB* has mild effects on the efficiency of conjugation but increases the host SOS response by up to six-fold [179]. PsiB interacts directly with RecA, thereby inhibiting several activities, such as DNA binding, LexA cleavage, and the strand exchange reaction [192,193]. SOS response inhibition by PsiB is even more potent in the presence of the SSB_C_ protein that coats the ssDNA. PsiB is well conserved among conjugative plasmids and is considered important for the early steps of plasmid establishment in the recipient, consistent with its early production in the transconjugant cells [57,191].

#### 3.3.3. Roles of the Host and Plasmidic SSB Proteins in Plasmid Establishment

Upon entry into the recipient cell, the transferred ssDNA plasmid is coated by the host SSB_C_ protein. SSB_C_ is a universally conserved essential protein that binds non-specifically to ssDNA. It is involved in various mechanisms, including DNA replication, repair, and recombination; SOS induction; and other DNA metabolic processes [194,195]. Upon binding, SSB protects the ssDNA against enzymatic degradation and increases the processivity of DNA polymerases II and III [196] during the replication reaction that converts the ssDNA strand into the dsDNA helix. The rapid recruitment of the host SSB_C_ protein to the transferred ssDNA has recently been visualized by fluorescence microscopy [197]. This work revealed that SSB_C_ proteins are rapidly recruited to the ssDNA that penetrates the recipient cell, presumably protecting it and facilitating its processing (Figure 1, step a). Interestingly, the F plasmid, as in many other conjugative plasmids, encodes its own SSB_P_ protein, which is homologous to the *E. coli* SSB_C_ [198]. One might then ask, what is the benefit gained by conjugative plasmids encoding their own SSB_P_, and what is the specific function of SSB_P_ compared with the chromosomal SSB_C_?

SSB_P_ binds ssDNA non-specifically, and SSB_P_ of different incompatibility groups (IncF, IncI, IncY, Inc9, IncT, and IncB/O) can partially complement conditional mutations of the *E. coli ssb_C_* gene [197,199,200,201]. Although the expression of the F plasmid-encoded SSB_P_ protein in *trans* enables the growth of the *ssb_C_* deletion mutant, complemented mutants exhibit some filamentation and growth rate reduction [202]. Moreover, a reduced affinity to ssDNA is observed for the F plasmid SSB_P_ in comparison with the *E. coli* SSB_C,_ and the F plasmid SSB_P_ cannot stimulate the reaction of DNA synthesis by DNA polymerase III in vitro [203]. Sequence alignment revealed that SSB_P_ proteins that complement the *E. coli ssb_C_* mutant share high homology only with the N-terminal part of *E. coli* SSB_C_ [203]. The SSB_C_ N-terminal region contains the domains for ssDNA binding and monomer–monomer interactions to cooperatively maintain the binding of the tetrameric structure of SSB_C_ to ssDNA. This structural conservation would explain the ability of SSB_P_ to bind ssDNA. However, the C-terminal domains of SSB_P_ are much more homologous to each other than to SSB_C_ [203]. However, as this domain interacts with partner proteins that constitute the SSB_C_ interactome, one possibility would be that the interactome of SSB_P_ and the reaction it is involved in might be different from that of SSB_C_.

Not all SSB_P_ proteins have been shown to complement *E. coli ssb_C_* mutants [199]. The latter study found that the inability of SSB_P_ from IncP-like RK2 to complement the *E. coli ssb_1_* mutation could be attributed to *ssb_P_* gene repression by the RK2 *kor* genes and that a derepressed plasmid indeed complemented the thermosensitive growth of *E. coli ssb_1_* mutations [204]. It is thus reasonable to consider the possibility that abundant SSB_P_ could also complement the *ssb_C_* mutant.

To date, the function of the F plasmid SSB_P_ in the context of conjugation is still unclear. The SSB_P_ protein could contribute to the protection of the transferred ssDNA by inhibiting enzymatic degradation or the recruitment of inhibiting proteins of the host. However, its expression timing instead supports the idea that SSB_P_ could be involved in the complementary strand synthesis of the transferred DNA or could simply increase the pool of available single-strand binding protein, which is required for the first cycle of vegetative replication of the plasmid.

### 3.4. Plasmid Maintenance: Replication and Segregation

Maintenance of the newly acquired dsDNA conjugative plasmid in the recipient cell lineage depends on two main active mechanisms: plasmid replication and segregation of the plasmid copies into daughter cells over generations.

The mechanisms of plasmid DNA replication in bacteria have been extensively studied and are the focus of well-referenced reviews [123,128,205,206,207,208]. Here, we stress the role of replication in conjugation host-range specificity. The F plasmid is efficiently transferred to *E. coli* and relatively close enterobacteria, while no transconjugant can be recovered after mating with more distant bacteria, such as *Vibrio* or *Pseudomonas*. As early as 1982, host range restriction was attributed to the plasmid’s inability to replicate in the recipient bacteria rather than to the inefficiency of plasmid transfer *per se* [188]. This was demonstrated by showing that mobilizable plasmids containing a cis-acting origin of transfer of the F plasmid and an origin of replication that is active in the tested recipient can then be transferred by the F conjugation machinery in *Pseudomonas* [188], *V. cholerae* [209], and yeast [98]. The same approach was employed to show that pCTX-M3, an IncI-like plasmid, was able to use its conjugation machinery to transfer a mobilizable plasmid to host recipients in which pCTX-M3 does not replicate [32]. The failure to replicate the F plasmid in *P. aeruginosa* comes in part from an inability of the plasmid replication protein RepE in the RepFIA replicon in complex with host DnaA-*oriS* to form a stable interaction with the host helicase DnaB [210]. In contrast, broad host range plasmids such as RK2 regulate their maintenance by modulating alternative strategies of replication depending on the host [211]. These findings indicate that the host range of the conjugation machinery and replication origins belonging to the same plasmid differ and can be mechanistically uncoupled. The specificity of narrow host range plasmids appears to be limited by the specificity of their replicon rather than by their transfer range.

Maintenance of the newly acquired dsDNA plasmid also requires the segregation of plasmid copies into the daughter cells during transconjugant cell division. To do so, low copy number plasmids encode active partition systems, of which the mechanism, biological functions, and conservation have been extensively reviewed [212,213,214,215,216]. In the context of conjugation, it is worth mentioning another maintenance strategy that involves the integration of the newly acquired conjugative DNA into the chromosome. Chromosome integration ensures the stable inheritance of the conjugative elements by vertical gene transfer over generations. The first characterized example was again the F plasmid, which uses insertion sequences (IS) and RecA-dependent homologous recombination to integrate into the *E. coli* genome [217]. The integrated plasmid can still initiate the conjugative transfer of the whole chromosome of the resulting “Hfr” strain (high frequency of recombination). This process of transfer is both progressive and oriented, and the order in which chromosomic genes are transferred depends on the position where the F plasmid was integrated. Alternatively, the integrated F plasmid can be excised out of the chromosome and recover its original autonomous form [218]. Chromosome integration is widely used by mobile DNA, such as ICE (Integrative Conjugative Elements), transposons, and phages, in both Gram-negative and Gram-positive bacteria; all of these systems have their particularities, especially regarding the recombination systems used for their integration/excision [219,220].

### 3.5. Phenotypic Conversion of the Transconjugant

The expression of genes carried by the newly acquired genetic element results in the phenotypic conversion of the recipient cell into a transconjugant that exhibits additional metabolic properties (Figure 1, step d). The expression of plasmid genes involved in DNA transfer converts the transconjugant into a new donor that is able to further transfer the plasmid to the population, thus accounting for the exponential rate of conjugative plasmid dissemination (see *tra* gene expression, Section 2.1). Not all genes encoded within the *tra* region of the plasmid backbone are directly involved in the process of DNA transfer. Indeed, conjugative plasmids often carry immunity (or exclusion) gene systems that are widespread in Gram-negative and Gram-positive organisms [90,221,222,223,224,225,226,227,228]. These immunity systems limit the ability of plasmid-carrying cells to serve as a recipient for the same plasmid [42,54,62,229]. Preventing self-mating by surface exclusion is thought to avoid the metabolic cost and potential cell death associated with repeated plasmid transfer but also to be important for plasmid stability and evolution [62,229].

The F plasmid immunity system relies on two factors—TraT and TraS proteins, neither of which is required for F-pilus synthesis or DNA transfer [56,230,231,232]. These two exclusion factors work at different levels. TraT is an abundant outer membrane protein that is thought to span the cell surface [230,231]. TraT production inhibits the formation of stable mating aggregates, presumably by interfering with the interaction between the pilus and recipient surface receptors. Consistent with this idea, it was reported that TraT interacts with OmpA, further suggesting that it could compete with TraN, which is key to mating pair stabilization [85,86,233]. TraS is an inner membrane protein, the production of which only slightly reduced the aggregation of mating populations but reduced DNA transfer frequencies by 100–200-fold. It is thus proposed that TraS acts by preventing DNA transfer when stable mating aggregates have already formed [230,231,234]. In F-like plasmids, it is proposed that TraS interacts with TraG to achieve the entry exclusion process [89,90]. For these reasons, TraS is referred to as an entry exclusion protein (Eex), and TraT as a surface exclusion protein (Sfx). Like other *tra* genes, *traS* and *traT* expression is controlled by TraJ, implying that transconjugant cells concomitantly acquire plasmid transfer ability and immunity to self-transfer during the phenotypic conversion.

In addition to genes located on the plasmid backbone, conjugative plasmids may carry additional genes that are not directly involved in conjugation but in a variety of biological functions, such as virulence, biofilm formation, symbiotic lifestyle, membrane trafficking, resistance to heavy metals, and, most importantly, resistance to antibiotics. Acquisition of these metabolic functions potentially facilitates bacterial adaptation and survival in changing environments and makes conjugation a major driver of the evolution of bacterial genomes. The successful maintenance of conjugative elements in bacterial populations shows that this selective advantage compensates for the metabolic burden associated with the metabolism of the newly acquired genetic information (highjacking of the host replication, transcription, and translation machineries) [235]. The most prominent example is the acquisition of conjugative drug resistance plasmids, which enable bacterial proliferation in microbial communities that contain antibiotic-producing organisms or in antibiotic-polluted and clinical environments. Indeed, analysis of commensal, environmental, and clinical antibiotic-resistant pathogenic strains revealed a multitude of conjugative plasmids that carry one or more genes for resistance to most, if not all, classes of antibiotics currently used in clinical treatments. Conjugation is considered to be the most widespread intra- and interspecies resistance transfer mechanism, accounting for 80% of acquired resistance [236].

## 4. Conjugation in Natural Habitats: The Example of Bacterial Biofilms

Gene transfer by conjugation is known to contribute to the genetic dynamics of bacterial populations living in a variety of environments, including the soil, on plant surfaces, and in water and sewage, as well as in bacterial communities associated with plant or animal hosts [4]. Bacteria are generally considered to be planktonic unicellular organisms, yet in natural and clinical environments, they often live in complex structures called biofilms. Biofilms shelter bacteria from external hazards but have also been proposed to offer a niche that facilitates the dissemination of drug resistance determinants by conjugation. Below, we review our current understanding of the interplay between biofilms and bacterial conjugation.

### 4.1. The Biofilm as a Niche Promoting Bacterial Conjugation

In natural environments, bacteria predominantly live in spatially structured communities termed biofilms, in which a self-produced extracellular matrix holds the cells together [237]. Planktonic cells that initiate biofilm formation can adhere to a living or inert surface (surface-attached biofilms) or can be present at the air–liquid interface as a free-floating community (pellicle). Bacterial biofilms are found in virtually every ecosystem on Earth, from aquatic systems (sludge, rocks, and wastewater) to terrestrial environments (rhizosphere) and human organisms (skin, intestinal, urogenital, and respiratory tracts). Furthermore, biofilms are associated with persistent and severe infections because of their ability to colonize medical devices and implants. Indeed, biofilm structures offer protection against hostile environments and, more worryingly, against antibiotic treatments. The biofilm architecture depends on the bacterial species, the surface colonized, and environmental conditions, but this lifestyle is characterized by the production of extracellular polymeric substances (EPSs) mainly composed of polysaccharides, proteins, lipids, and extracellular DNA. EPS matrix production is dynamic and continuous and mediates the formation of the biofilm architecture in which aggregates of microorganisms are trapped.

Many studies on bacterial conjugation have shown that plasmid transfer can occur in both natural and artificial biofilms, from the aquatic environment [238], phytosphere [239], animal and human hosts [240,241], or reactor-associated biofilms [242,243]. These papers studied plasmid transfer at the population level and mainly relied on limited cultivation-based assays that probably underestimate the extent of conjugation in natural biofilms. Further works have studied conjugative plasmids expressing fluorescent markers, allowing the direct in situ visualization of donors, recipients, and transconjugants at the single-cell level within several types of biofilms formed at the liquid–air surface [244], on a semisolid agar surface [245,246,247], on a filter [248], in flow chambers [246,249,250], or in a reactor [251].

The biofilm environment provides a high cell density and close cell-to-cell proximity that may facilitate HGT through bacterial conjugation. In line with the view that biofilm is a hot spot niche of conjugation, several studies have shown that the frequency of plasmid transfer is higher in the biofilm than in the planktonic mode of growth [252,253,254,255]. Although the biofilm appears to be a favorable environment for HGT by conjugation, many studies at the single-cell level have reported limited plasmid propagation inside a preformed established biofilm beyond the contact zone between the donors and recipients [256]. These observations are directly linked to the complex spatial structure of the biofilm, which might have a critical impact on horizontal plasmid spread within a biofilm.

### 4.2. Impact of the Biofilm Structure on Conjugation

The biofilm is a complex structure and sometimes composed of mixed bacterial species. The matrix shapes the spatial organization by clustering cells in microcolonies in an architecture characterized by nonuniform cell arrangements, open channels, pores, cavities, and different layers of living cells [257,258]. Such an organization determines the formation of cell clusters/aggregates and could therefore influence the efficiency of conjugative transfer [244]. This possibility can be addressed using microscopy to analyze the distribution of cells that are active in conjugation within biofilms. To date, most studies have investigated the spread of the GFP-tagged *Pseudomonas putida* TOL plasmid. Analysis of *P. putida* recipient biofilms established in flow chambers revealed that transconjugants appeared at the top surface of the biofilm but not in the deeper layers, reflecting a limited invasion of the transferred plasmid [246,250]. On an agar plate, transconjugants only appeared at the contact zone between the TOL plasmid donors and the recipient colonies [245,246,259]. Limited transfer to the outer layers of the biofilm was also observed for IncF, IncI, and IncW plasmids in *E. coli* [247]. It has been proposed that transfer can only be efficient over a short period between metabolically active cells growing at the donor–recipient interface [246,260]. Consistently, it was shown that plasmid invasion stops in non-dividing cells [259].

Within biofilm structures, chemical gradients of oxygen, nutrients, temperature, and pH create microenvironments that influence the metabolic activity of bacterial cells [261]. This results in physiological heterogeneity between the cells that surround the border of the biofilm and those that are embedded deep inside. Furthermore, variations in the spontaneous mutation frequency within parts of the biofilm result in the emergence of variant subpopulations with genetic heterogeneity [262]. Whether and how these factors impact the spatial pattern of conjugation within biofilms remain unknown.

### 4.3. Impact of Conjugative Plasmids on Biofilm Formation

Several studies have investigated the implication of the presence of conjugative plasmids from diverse incompatibility groups for biofilm formation ability [263,264,265,266,267]. To initiate biofilm formation, planktonic cells produce cell appendages like flagella and adhesion factors such as type IV pili and type 1 and curli fimbriae [268,269]. Genes coding for these types of accessory factors that promote attachment to biotic or abiotic surfaces are often found in conjugative plasmids, resulting in increased host biofilm formation [253]. Examples include type 3 fimbriae of the IncX1 plasmid pOLA52 [270], non-conjugative type IV pili of the IncI1 plasmid pSERB1 [271], or pilus-like structure and surface adhesins of the *Enterococcus faecalis* plasmids pBEE99 and pCF10, respectively [272,273,274]. In 2001, Ghigo made the unexpected observation that the conjugative pilus itself of a derepressed F plasmid can promote the biofilm of *E. coli* cells that are initially unable to form such a structure and revealed that the pilin TraA is the main adhesion factor that induces biofilm formation [263]. Microscopic structure analysis of derepressed IncF plasmid R1*drd*19- and F-carrying *E. coli* biofilm showed the rapid formation of a dense and mature 3D mushroom-type biofilm similar to the *P. aeruginosa* biofilm architecture [275]. The formation of this peculiar architecture and the biofilm maturation generated by derepressed plasmids override the need for cell surface appendages such as flagella, type 1 fimbriae, Ag43, or curli, which are essential to *E. coli* biofilm [275]. In contrast, maturation of the 3D mushroom-type biofilm structure depends on curli production, induced in *E. coli* by the natural F plasmid, which does not constitutively express F-pili [276]. The presence of the plasmid R1*drd*19 also increases *E. coli* biofilm formation by decreasing the motility and increasing the level of quorum-sensing inducer AI-2 [277,278], and the IncP-9 TOL plasmid in *P. putida* increases the production of extracellular DNA known to play a role in the structure of the biofilm [279] and thus the biofilm formation capacity [280]. However, the genetic mechanism by which conjugative plasmids increase biofilm formation has not been elucidated, but the presence of repressed or natural conjugative plasmids affects the global host chromosomal gene expression [276,278].

While the role of the conjugative pili has been mainly studied in *E. coli* biofilms, one may speculate that their impact differs depending on the host. Indeed, Røder et al. observed that the conjugative pili of the IncP-1 plasmid pKJK5 reduced the surface attachment of *P. putida* by increasing cell–cell adhesion, resulting in reduced biofilm formation [281]. Further investigation will be necessary to decipher the complex interconnections between the conjugative plasmid and biofilm formation.

### 4.4. Influence of Antibiotic Treatment on Conjugation within Biofilms

Interactions between conjugation and biofilms have been proposed to promote both community-building and gene transfer. This synergic interaction raises serious questions about the contribution of HGT to the evolution and adaptation of biofilm-forming pathogens. Because of the increase in antibiotic-resistant infections, recent investigations have aimed to provide a new understanding of biofilm responses to antimicrobial treatments.

Subminimal inhibitory concentrations (sub-MICs) of aminoglycosides enhance the biofilm biomass of the *P. aeruginosa* strain PAO1 and clinical isolates of *E. coli* through the response regulator Arr, a predicted phosphodiesterase that alters cyclic di-guanosine monophosphate (c-di-GMP) levels [282]. Linares et al. further demonstrated that, in addition to aminoglycosides, sub-MICs of tetracycline and norfloxacin also increase the formation of *P. aeruginosa* biofilm. However, no clear causal factors were identified [283]. Interestingly, a combination of tetracycline and cephradine has a synergistic effect on the biofilm formation of a mixed culture of *E. coli* and *P. aeruginosa* [284]. Tetracycline also promotes the biofilm formation of the pathogen *Acinetobacter baumannii*, and whole-genome sequence analysis revealed an increase in the rate of mutations such as SNPs, as well as insertions and deletions, under subinhibitory drug exposition [285]. In addition to the accumulation of genotypic variation, biofilm treatment with a low level of antibiotics produces changes in the gene expression profile, some of which may be linked to increased biofilm formation [283,285,286].

Recently, Diaz-Pascual et al. investigated *Vibrio cholerae* biofilm at the community scale using a single-cell imaging system, revealing changes in biofilm dynamics and architecture in response to antibiotic treatment [287]. After tetracycline exposure, they observed modifications in biofilm architecture and cell morphology, including a 2.5-fold increase in cell volume and a 29% decrease in cell density. This cell density decrease reflected a considerable alteration of the multicellular arrangement and the breakdown of the matrix within the biofilm. Furthermore, biofilms became susceptible to the colonization of their interior by new cells, and the colonizer population increased until they invaded the resident biofilm. Clearly, a sublethal dose of antibiotics influences the biofilm lifestyle, inducing significant modifications to the entire population. The biofilm matrix forms a shield to prevent the penetration and diffusion of antimicrobials, and the increase in biofilm formation in response to antibiotics, illustrated by an enhancement of the biofilm matrix, seems to be a defense mechanism of the bacterial community.

In parallel, some antibiotics have also been recognized as signaling molecules that increase conjugative transfer [288,289,290,291,292,293,294,295,296]. Interestingly, when conjugation occurs in an antibiotic-free environment with a donor strain that is pretreated with subconcentrations of antibiotics, the conjugation frequency increases significantly [297,298]. The mechanisms by which antibiotics affect plasmid transfer remain unclear. In the literature, it is proposed that sub-MIC antibiotic treatment enhances the frequency of conjugation through the upregulation of *tra* gene expression in donors [291,297,298,299]. However, in many studies, the increase in conjugation frequency was evaluated using an antibiotic whose resistance gene is carried by the tested conjugative plasmid itself [288,290,291,293,294,299,300]. This approach makes it difficult to distinguish between the selection bias induced by the antibiotic in the mating population and the actual effect on conjugation frequencies. Two studies instead support that antibiotics primarily act through the differential selection of the donor, recipient, and transconjugant once gene transfer has occurred, rather than stimulating conjugation *per se* [197,301]. By using living cell microscopy, they were able to visualize conjugation dynamics in real time. Nolivos et al. showed that the transfer frequency of an F plasmid harboring the gene for tetracycline resistance was not increased by the presence of tetracycline. Lopatkin et al. also demonstrated that antibiotics from six major classes had no effect on the conjugation efficiency of plasmids from five different incompatibility groups. These reports suggested that the direct contribution of antibiotics to gene transfer has been overestimated and proposed that antibiotics may act only as post-transfer selection drivers, favoring the growth of transconjugants over recipients. Although a number of studies have advanced the potential stimulating effect of antibiotics on conjugation, their real impact needs to be further explored, and antibiotics with different modes of action must be tested.

Undoubtedly, antibiotics play a role that must not be overlooked in the emergence of new multi-resistant pathogenic strains. It is troubling that antibiotic treatments amplify biofilm formation, increasing the difficulty in healing biofilm-associated infections. Antibiotics not only induce biofilm formation but also improve gene transfer within the community. Furthermore, profound changes induced by antibiotics allow for the invasion of the biofilm by external microorganisms [287]. As biofilms are suitable environments for conjugative transfer, we can easily imagine that antibiotics could potentiate the invasion by a potential donor that harbors gene resistance and that its dissemination within the biofilm could act as a synergistic factor instead of an antagonist one. Microfluidic technology represents a promising method to investigate, in real-time and without disrupting biofilm structure, the dynamics of conjugation within communities. Recent studies have used microfluidic devices combined with confocal microscopy to monitor real-time plasmid RP4 transfer in mixed *P. putida* and *E. coli* biofilms and in activated sludge [302]. They were able to show that the structure and composition of the biofilm could modulate gene transfer routes. Indeed, in *E. coli* biofilms, the explosive spread of transconjugants illustrated the significant role of plasmid transfer, while in the sludge community, vertical gene transfer was more predominant. Using these advanced techniques, it is now more necessary to understand how antibiotics can influence gene dissemination within these complex structures.

## 5. Conclusions

Our current knowledge of the sequence of reactions required for plasmid conjugation is well documented, especially for model plasmids such as the F factor, but also for other plasmids like RP4, R388, or pTi. The combination of genetic and biochemistry approaches has allowed the function of key Tra proteins in these reactions to be described. However, even for these systems, the mechanistic functions of most Tra proteins remain elusive. They have mainly been described in terms of their essentiality for mating pair formation or stabilization, DNA transfer, and immunity, and a further understanding of their activity at the molecular scale is lacking. Furthermore, as emphasized in this review, a number of major fundamental questions remains, such as the pilus’ ability to transport DNA during distant transfer, the existence and nature of a potential signal that is triggered by mating pair formation that would activate conjugation, or the role of the leading genes in the early steps of plasmid establishment, for instance. Because of its intimate connection with the dissemination of drug resistance, conjugation has reemerged as the focus of a global research effort. Modern experimental approaches should help to gain new insights into the mechanism of conjugation at the molecular and cellular scales, as well as those regarding the extent of conjugation in natural bacterial communities and its impact on the dissemination of bacterial metabolic traits.

## Figures and Tables

**Figure 1 genes-11-01239-f001:**
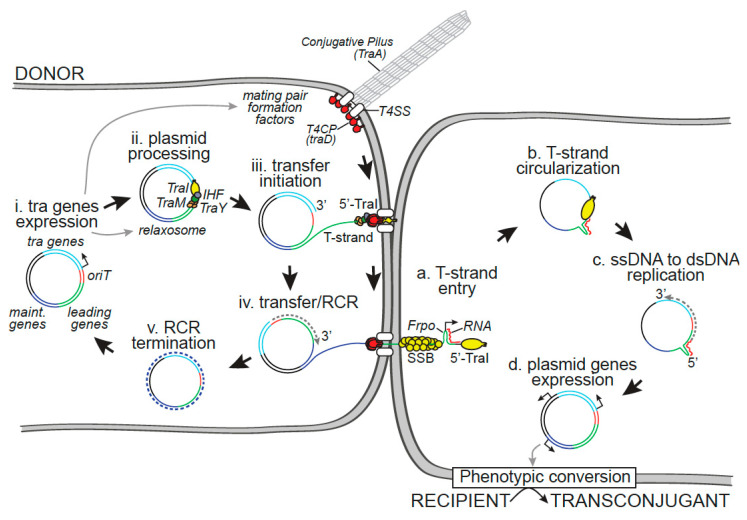
Schematic diagram of the life cycle of the F plasmid during conjugational transfer from the donor to the recipient cell. This F plasmid backbone is composed of the *tra* regions encoding all genes involved in conjugational transfer (light blue); the origin of transfer *oriT* (red); the leading region (green), which is the first to be transferred into the recipient cell; and the maintenance region (dark blue) involved in plasmid replication and partition. (**i**) The initiation of conjugation requires the expression of the *tra* genes. Some of the produced Tra proteins form the T4SS and the conjugative pilus that will recruit the recipient cell and mediate mating pair stabilization. (**ii**) Other Tra proteins constitute the relaxosome (TraI, TraM, and TraY), which, in combination with the integration host factor (IHF), bind to the *oriT* and prepare the plasmid for transfer by inducing the nicking reaction by the TraI relaxase. (**iii**) Interaction between the relaxosome and the Type IV Coupling Protein (T4CP) initiates the transfer of the T-strand by the T4SS. (**iv**, **v**) Transfer of the TraI-bound T-strand in the recipient is concomitant with the conversion of the ssDNA into dsDNA by Rolling Circle Replication (RCR) in the donor. (**a**) Upon entry into the recipient, the ssDNA T-strand is coated by the host chromosomal SSB, and the single-stranded promotor Frpo adopts a stem-loop structure recognized by the host RNA polymerase to initiate the synthesis of RNA primers. (**b**) TraI performs the circularization of the fully internalized T-strand. (**c**) The RNA–DNA duplex is recognized by the host DNA polymerase to initiate the complementary strand synthesis reaction. (**d**) Once the conversion of the ssDNA plasmid into dsDNA is completed, plasmid gene expression results in the phenotypic conversion of the recipient cell into a transconjugant cell.

**Figure 2 genes-11-01239-f002:**
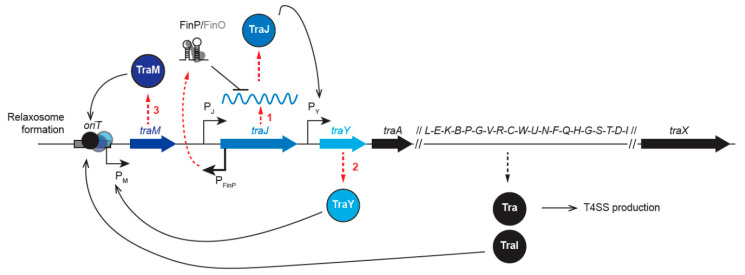
Activation cascade of *tra* gene expression. The P_J_ promoter first drives *traJ* expression (1). Translated TraJ protein binds to the P_Y_ promoter to notably produce TraY, which activates the P_M_ promoter (2), other Tra proteins constituting the T4SS, and the relaxase TraI. Once produced (3), TraM autoregulates its own expression through the P_M_ promoter and, in combination with TraY and TraI, forms the relaxosome bound to *oriT*. The activation of this regulatory cascade is modulated by the FinP/FinO complex, which represses the translation of TraJ at the post-transcriptional level. Dotted red arrows illustrate the transcription–translation process.

**Table 1 genes-11-01239-t001:** Description of Tra proteins. Proteins are presented following the order of the corresponding *tra* genes in the *tra* region of the F plasmid. The proposed function, the description of their biological activity, the subcellular localization (IM: Inner membrane; OM: Outer membrane; C: Cytoplasm; P: Periplasm), and the homologs in RP4, pTI, or R388 plasmids are shown.

Protein	Proposed Function	Description	Localization	Homolog	Reference
TraM	Relaxosome	*oriT* binding, TraI stimulation,	C		[17,52,55,56]
		Interaction with TraD			
TraJ	Regulation	Transcription factor (anti-silencer/activator of P_Y_)	C		[17,55,57]
TraY	Relaxosome Regulation	*oriT* binding, Transcription factor (activator of P_M_)	C		[17,52,55,57]
TraA	Pilin	Major subunit of the pilus	IM	VirB2 (pTi)	[52,55,56]
				TrbC (RP4)	
TraL	Pilus assembly	Pilus assembly	OM	VirB3 (pTi)	[52,53,55,57,58]
				TrbD (RP4)	
TraE	Pilus assembly	Pilus assembly	IM/P	VirB5 (pTi)	[52,53,55]
TraK	Pilus assembly	Cell envelope-spanning channel	IM/P	VirB9 (pTi)	[52,53,55,56]
TraB	Pilus extension	Cell envelope-spanning channel	IM	VirB10 (pTi)	[52,53,55,56,58]
				TrbI (RP4)	
TraP	Pilus extension	Extended pilus stabilization	IM		[55,57]
TraG	Pilus assembly	Pilus tip assembly	IM	VirB6/VirB8 (pTi)	[52,53,55,57,59]
	Mating pair stabilization	Stabilization via C-terminal Interaction with TraN,			
	Exclusion	Interaction with TraS			
TraV	Pilus extension	Lipoprotein	OM/P	VirB7 (pTi)	[52,53,55,56]
TraR	Regulation	Transcription regulator by binding to RNA polymerase	C		[58,60]
TraC	Pilus assembly	NTPase	IM	VirB4 (pTi)	[52,53,55,56]
				TrbE (RP4)	
TraW	Pilus extension	Pilus synthesis	P		[52,53,56,57,61]
TraU	DNA transfer	DNA transfer	P		[52,55,56]
TraN	Mating pair stabilization	Stabilization of OmpA and Lps binding	OM		[52,55,56]
	Exclusion system	Interaction with TraG			
TraF	Pilus extension	Disulfide bonds for T4SS assembly	P		[52,53,55,56]
TraQ	Pilin maturation	Chaperone-like	IM		[55,56,57]
TraH	Pilus extension	Interaction with TraF and TraU	P		[52,55]
TraG	Pilus assembly	Pilus tip assembly	IM	VirB6/VirB8 (pTi)	[52,55,57,59]
	Mating pair stabilization	Stabilization via C-terminal Interaction with TraN,			
	Exclusion	Interaction with TraS			
TraS	Entry Exclusion (Eex)	Interaction with TraG	IM		[55,56]
TraT	Surface exclusion (Sfx)	Disaggregation of mating pair after DNA transfer, Interferes with TraN–OmpA interaction	OM		[55,56,62,63]
TraD	T4CP	Coupling protein/DNA dependent ATPase Interaction with TraM	IM	VirD4 (pTi)	[55,56,57,64]
				TraG (RP4)	
				TrwB (R388)	
TraI	Relaxosome	Relaxase, transesterase and helicase	C	VirD2 (pTi)	[55,57,65,66]
				TrwC (R388)	
TraX	Pilin maturation	N-terminal acetylase	IM	TrbP (RP4)	[55,56,57](1)(3)(8)

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
