# Peer review of "Plasmid Transfer by Conjugation in Gram-Negative Bacteria: From the Cellular to the Community Level"

_genes, 2020, doi:10.3390/genes11111239_

Round 1
Reviewer 1 Report
This article is a review of plasmid transfer in bacteria (it claims to be a review of conjugation in general, but really does not deal with ICE, conjugative transposons, mobile genomic islands and so on). It focuses on the well known F plasmid, but does give some interesting asides about other systems that do provide useful insights. However, I found that there was little in here that has not already been covered in previous reviews like those of Frost or Glover on F, and others on plasmids like RK2/RP4, and Pappas and colleagues on Agrobacterium. Then there is last section on biofilm plasmid transfer which is not without interest, but really does not flow well from the first part. It would make far more sense to make that part the subject of a separate minireview, and to expand and update the more technical aspects of plasmid transfer and its regulation. As it stands much of this review is too superficial.
More specifically:
Sections 2.1.1 needs more breadth including the various types of regulation of plasmid transfer (and ICE transfer) in rhizobia, and in gram positive bacteria with their "pheromone" systems.
Sections 2.1.2 -should address here why the transfer rates of RP4/RK2 type plasmids are so high. Are they super spreaders ? Is it their rigid pili (rigid pili do not get enough discussion)
Section 3.2. eg. line 336 and on. Need to address the fact that most restriction systems recognize and target ds DNA, and to expand this section to explain how and when ssDNA would get targeted. Presumably methylation of target sites would happen during second strand synthesis.
In terms of regulation of plasmid transfer, one logical way for this to be accomplished is to have potential recipient cells up-regulate transfer (as happens in gram positive plasmids like pAD1, and also in some rhizobial plasmids through a compicated QS system); the apparent lack of recipient regulation in other systems deserves serious discussion.
The MS is poorly edited and English usage is substandard. The whole paper needs complete revision and editing for grammar by a native English speaker or professional editing service. Below, by line number, I list just some of the instances of incorrect usage - there are many many more.
By line number
29. Coined THE term (needs definite article)
54. Is "chapter" really the appropriate word in this context
107. Agrocinopine should be singular, and anyway, this is not a general feature of all Ti plasmids, since different opines regulate the transfer of the plasmids in quite different ways. There are many reviews on this that the authors should read.
157-158. This is false. A pilus is not always required. Many gram positive systems do not use a pilus, for example.
165. Processed, not maturated (which is not a word)
195. it was first proposed (not early proposed)
383 last word, plasmid should be singular.
387. encode OR code for. NEVER encode for, that is tautological
492. close, not closed
592. Plasmid, not plasmids transfer
And so on.
Author Response
REVIEWER1
Comments and Suggestions for Authors
This article is a review of plasmid transfer in bacteria (it claims to be a review of conjugation in general, but really does not deal with ICE, conjugative transposons, mobile genomic islands and so on). It focuses on the well known F plasmid, but does give some interesting asides about other systems that do provide useful insights. However, I found that there was little in here that has not already been covered in previous reviews like those of Frost or Glover on F, and others on plasmids like RK2/RP4, and Pappas and colleagues on Agrobacterium. Then there is last section on biofilm plasmid transfer which is not without interest, but really does not flow well from the first part. It would make far more sense to make that part the subject of a separate minireview, and to expand and update the more technical aspects of plasmid transfer and its regulation. As it stands much of this review is too superficial.
We thank this reviewer for his/her comments and suggestions, which helped improving our manuscript. This reviewer is right that our review focuses on conjugation plasmid found in Gram-negative bacteria. This is now clearly stated in the introduction (line 63) as well as in the indexation keywords. Besides, we are sorry that the mechanism and the regulation of conjugation could not be addressed comprehensively for all organisms. Our first objective was to cover the key steps of transfer using the F plasmid as an example. However, we thank this reviewer for appreciating that we managed to provide some “useful insights”, despite the fact that some interesting examples have not been included in this review.
More specifically:
Sections 2.1.1 needs more breadth including the various types of regulation of plasmid transfer (and ICE transfer) in rhizobia, and in gram positive bacteria with their "pheromone" systems.
It is now indicated that this section only discusses other Gram-negative conjugation systems. We have decided to focus on a few illustrative examples and we are sorry that the diversity of transfer regulation could not be addressed comprehensively (or in rhizobia specifically).
Sections 2.1.2 -should address here why the transfer rates of RP4/RK2 type plasmids are so high. Are they super spreaders ? Is it their rigid pili (rigid pili do not get enough discussion)
The natural RP4 plasmid is not considered as a superspreader per se, since no genetic mutation has been reported to be responsible for its transfer rate. However, we cite an article reporting that a superspreader mutant of the pESBL plasmid is indeed associated with the overproduction of pili that do not look like long filamentous F-like pili.
Section 3.2. eg. line 336 and on. Need to address the fact that most restriction systems recognize and target ds DNA, and to expand this section to explain how and when ssDNA would get targeted. Presumably methylation of target sites would happen during second strand synthesis.
This reviewer it right. We have clarified this point line 951-954.
In terms of regulation of plasmid transfer, one logical way for this to be accomplished is to have potential recipient cells up-regulate transfer (as happens in gram positive plasmids like pAD1, and also in some rhizobial plasmids through a compicated QS system); the apparent lack of recipient regulation in other systems deserves serious discussion.
The lack of regulation by recipient factors in the case of the F plasmid is discussed in sections 2.2.3. Conversely, the existence of transfer regulation by the recipient through quorum sensing is mentioned in section 2.1.1.
The MS is poorly edited and English usage is substandard. The whole paper needs complete revision and editing for grammar by a native English speaker or professional editing service. Below, by line number, I list just some of the instances of incorrect usage - there are many many more.
The revised manuscript has been edited by MDPI professional English editing service and now includes substantial rewriting and corrections. We sincerely apologize that the English writing was poor. We thank this reviewer for evaluating our manuscript in-depth nonetheless.
By line number
- Coined THE term (needs definite article)
- Is "chapter" really the appropriate word in this context
“Chapter” has been replaced by “section”
- Agrocinopine should be singular, and anyway, this is not a general feature of all Ti plasmids, since different opines regulate the transfer of the plasmids in quite different ways. There are many reviews on this that the authors should read.
157-158. This is false. A pilus is not always required. Many gram positive systems do not use a pilus, for example.
- Processed, not maturated (which is not a word)
- it was first proposed (not early proposed)
383 last word, plasmid should be singular.
- encode OR code for. NEVER encode for, that is tautological
- close, not closed
- Plasmid, not plasmids transfer
All above corrections were made as suggested by the reviewer.
Once again, we thank this reviewer for assessing our work and we hope that we have addressed this reviewer’s concerns in full in the revised version of the manuscript.

Reviewer 2 Report
In the paper “Bacteria DNA conjugation: from the cellular to the community level” Virolle et al are describing the bacterial conjugation process in context of cell cycle. The idea of this review is very interesting. The structure of this paper is very well thought through. Authors have covered quite an extensive range of literature. However, the writing style and awkward phrasing makes this paper very challenging and even frustrating at times to read. A review dealing with such a complex topic should be written in much easier to understand way. Otherwise, the readers might not be willing to go through the entire manuscript. Admittedly the reviewer was considering that as well! Due to the difficulties with understanding what the authors were trying to say, the reviewer cannot clearly judge the quality of the scientific part of the manuscript. In addition the reviewer encourage the authors to check their references and make sure that they cite the papers accurately. Therefore, the reviewer recommends that the authors re-write the manuscript and resubmit it again to be considered. Below are the reviewer’s list of comments:
- As mentioned above, the major issue that the reviewer raises at this point is with the writing style of the authors. There are way to many examples within the manuscript to list them all here, therefore the reviewer will just highlight a few to demonstrate the problems.
The reviewer finds issue already in the first word of the title. It should be “Bacterial” rather than “Bacteria”
Line 240-243
The authors write, “…TraNpKpQIL would eventually mediate the OmpK36-receptor specificity…”.
What does it mean that protein is “mediating” specificity? The use of word “eventually” here is also questionable. Also the convention is to say “F plasmid” rather than “plasmid F”
- Authors also need to be more careful with their phrasing.
Line 267-271
The authors write “The initiation of conjugation requires the assembly and activity of a protein complex, the relaxosome, that processes the plasmid before DNA transfer (Figure 1, step ii). The relaxosome introduces a site- and strand-specific DNA cut (nick) at the nic site located in the origin of transfer oriT and is required for the extrusion of the single-stranded DNA to be transferred (T-strand)”
This sentence is very misleading. The only protein, which is doing any processing, is TraI. Not the relaxosome! The other proteins, which are commonly called accessory or auxiliary proteins, are only facilitating this process by altering the DNA structure.
Line 277-278
The authors write “TraM is a small protein…”
TraM forms a tetramer and therefore in fact is the biggest of all the auxiliary proteins. Even in the monomeric form TraM is bigger than TraY or IHF! Therefore, the reviewer does not understand why the authors felt the need to even make this comment?!
- As mentioned before the authors should be very careful when they cite papers
line 307-309
Authors write “Consistently, structure analysis by Cryo-EM shows that two relaxase molecules bind the oriT, one associated with the 5’ end that is in open transesterase conformation and one associated with the 3’ end that is in closed helicase conformation”
In the cited paper the conclusion about the two molecules of TraI binding to the oriT as well the two states of the relaxase were not based on cryoEM structure but biochemical assays!
- Line 325-326
Authors write “Such interaction has been speculated in the F system but remains elusive”
The reviewer would recommend reading paper: “An activation domain of plasmid R1 TraI protein delineates stages of gene transfer initiation” by Lang et al Mol. Microbiol. 2011 The authors in this paper demonstrate stimulation of relaxase activity of TraI in R1 system by protein-protein interaction with TraD, coupling protein.
- Line 388
Authors write “Mahendra and co-workers …”
In the section 4 of the manuscript the authors use a different way to refer to published work e.g. line 656 “Røder et al. indeed…”
This should be more uniform within the manuscript.
- Line 403
Authors write “…plasmid ssb (ssbP) and psiB genes are expressed early upon entry of the plasmid in the transconjugant bacteria…”
The DNA enter recipient cell/bacteria and not transconjugant.
- Line 404-405
Authors write, “This was demonstrated using lacZ fusions of the two genes inside the ColIb-P9 plasmid, and also by immunoassays concerning the F plasmid PsiB protein.”
Stating that lacZ fusions of two genes was used to demonstrate expression is not enough. This should be either clearly explained or not mentioned at all.
- Line 446
Authors write, “SSBC is an essential conserved protein…”
Essential for what? Conserved where?
- Line 448
Authors write, “SSB binding protects the ssDNA…”
In the view of the reviewer this is another example of the confusing style used by the authors. This sentence would be much clearer if they would write for example “Upon binding SSB protects the ssDNA…” It sound like a small change but makes a huge difference.
Another example
line 458
authors write “SSBP can bind ssDNA without specificity…”
What about “SSBp binds ssDNA nonspecifically…”
Another sentence with few issues.
Line 460 -461
Authors write “However, although producing plasmid F SSBP in trans enables the growth of ssbC deletion mutant, complemented mutant exhibit some filamentation, growth rate reduction.”
- It is either “However” or “although”
- “producing plasmid F SSBP” instead “expression of F plasmid encoded SSBP protein”
- “…growth rate reduction” ???? do the authors meant “…at reduced growth rate” ???
- Line 488-491
Authors write “Here we would like to stress the role of replication in conjugation host-range specificity. Indeed, the ability of the transferred plasmid to replicate in the recipient cells is considered as the primary determinant of plasmid host-range specificity and efficiency of conjugation.”
The two sentences here are basically stating the same thing!
- Line 508
Authors write ”… timely segregation…”
what does it mean that segregation is timely???
- Line 527
Authors write “…phenotypic conversion of the transconjugants into a novel type of cell that exhibits additional metabolic…”
However in the Figure 1 under the Box “Phenotypic conversion” they show conversion of the recipient into a transconjugant??? It will leave the reader very confused.
- Line 531
Authors write, “Other plasmid genes encoded by the tra region of the plasmid backbone are not directly involved in the transfer process per se.”
The authors need to specify what they mean by other plasmid encoded genes?? Do they want to say “not all the genes encoded within tra region are directly involved in the process of DNA transfer” ??? Very confusing
- Line 534
authors write, “These immunity systems limit the plasmid-carrying cell ability to serve…”
- Line 541
Authors write, “TraT production reduces the recipient…”
Reduces?? What does this mean?
- Line 551
Authors write “…plasmid transfer ability immunity…”
To many nouns!
- Line 594
Authors write “…researches studied…”
Very awkward phrasing.
- Line 661
Authors write “Dynamics of conjugation within biofilms need to be more investigated…”
This sounds more like a conclusion rather than an opening statement.
- Line 665
Authors write “…recent advances tried to bring…”
Another awkward phrasing
- Line 691-692
Authors write “In parallel, antibiotics have also been recognized as signaling molecules increasing conjugative transfer although not all antibiotics can potentiate the frequency of conjugation.”
The authors could write “…some antibiotics…” and then the whole part of the sentence coming after conjugative transfer is redundant.
- Line 706-708
Authors write “Lopatkin et al. also demonstrated that antibiotics, from six major classes, not significantly increase the conjugation efficiency of plasmids from five different incompatibility groups.”
“Not significantly increase” means that they have no effect!
What is “plasmid-carrying cell ability” ???
- Line 715
Authors write “ …strengthened the difficulty…”
Once again awkward phrasing.
- In the Figure 2 the authors draw the arrows of various genes but not up to scale. For a person who is not familiar with the gene organization this might suggest that the gene encoding the regulatory proteins are much larger than the ones encoding the actual T4SS components. This needs to be changed.
There are more issues related to the style or wording but the reviewer feels that this is too much to ask from him to point all of it out. Therefore, once again it is advised to re-write the manuscript and make it more legible, because in this form it is really very difficult to focus on scientific aspect of this manuscript.
Author Response
REVIEWER2
Comments and Suggestions for Authors
In the paper “Bacteria DNA conjugation: from the cellular to the community level” Virolle et al are describing the bacterial conjugation process in context of cell cycle. The idea of this review is very interesting. The structure of this paper is very well thought through. Authors have covered quite an extensive range of literature. However, the writing style and awkward phrasing makes this paper very challenging and even frustrating at times to read. A review dealing with such a complex topic should be written in much easier to understand way. Otherwise, the readers might not be willing to go through the entire manuscript. Admittedly the reviewer was considering that as well! Due to the difficulties with understanding what the authors were trying to say, the reviewer cannot clearly judge the quality of the scientific part of the manuscript. In addition the reviewer encourage the authors to check their references and make sure that they cite the papers accurately. Therefore, the reviewer recommends that the authors re-write the manuscript and resubmit it again to be considered. Below are the reviewer’s list of comments:
We thank this reviewer for the critical assessment of our work and for her/his constructive comments, which helped improve our manuscript. We sincerely apologize that the English writing was substandard, and we thank this reviewer for assessing our manuscript in-depth. The revised manuscript has been edited by MDPI professional English Editing service and now includes substantial rewriting and corrections. We also modified Figure 2 as suggested. We hope that we have addressed this reviewer’s concerns in full. Below, we provide a point-by-point answer to his concerns.
As mentioned above, the major issue that the reviewer raises at this point is with the writing style of the authors. There are way to many examples within the manuscript to list them all here, therefore the reviewer will just highlight a few to demonstrate the problems.
The reviewer finds issue already in the first word of the title. It should be “Bacterial” rather than “Bacteria”
This has been corrected.
Line 240-243
- The authors write, “…TraNpKpQILwould eventually mediate the OmpK36-receptor specificity…”. What does it mean that protein is “mediating” specificity? The use of word “eventually” here is also questionable. Also the convention is to say “F plasmid” rather than “plasmid F”
The term “mediate” has been replaced by “determine” and “eventually” has been removed. We also use “F plasmid” throughout.
- Authors also need to be more careful with their phrasing.
Line 267-271 The authors write “The initiation of conjugation requires the assembly and activity of a protein complex, the relaxosome, that processes the plasmid before DNA transfer (Figure 1, step ii). The relaxosome introduces a site- and strand-specific DNA cut (nick) at the nic site located in the origin of transfer oriT and is required for the extrusion of the single-stranded DNA to be transferred (T-strand)”. This sentence is very misleading. The only protein, which is doing any processing, is TraI. Not the relaxosome! The other proteins, which are commonly called accessory or auxiliary proteins, are only facilitating this process by altering the DNA structure.
We agree with this accurate comment. The text now reads “The initiation of conjugation requires the assembly and activity of a protein complex, the relaxosome, that allows the processing of the plasmid before DNA transfer (Figure 1, step ii). Plasmid processing involves a site- and strand-specific DNA cut (nick) at the nic site located in the origin of transfer (oriT) and the extrusion of the single-stranded DNA that will be transferred”
Line 277-278 The authors write “TraM is a small protein…”
TraM forms a tetramer and therefore in fact is the biggest of all the auxiliary proteins. Even in the monomeric form TraM is bigger than TraY or IHF! Therefore, the reviewer does not understand why the authors felt the need to even make this comment?!
This statement was useless indeed, and has been removed.
- As mentioned before the authors should be very careful when they cite papers
line 307-309
Authors write “Consistently, structure analysis by Cryo-EM shows that two relaxase molecules bind the oriT, one associated with the 5’ end that is in open transesterase conformation and one associated with the 3’ end that is in closed helicase conformation”
In the cited paper the conclusion about the two molecules of TraI binding to the oriT as well the two states of the relaxase were not based on cryoEM structure but biochemical assays!
This sentence has been corrected accordingly.
- Line 325-326
Authors write “Such interaction has been speculated in the F system but remains elusive”. The reviewer would recommend reading paper: “An activation domain of plasmid R1 TraI protein delineates stages of gene transfer initiation” by Lang et al Mol. Microbiol. 2011 The authors in this paper demonstrate stimulation of relaxase activity of TraI in R1 system by protein-protein interaction with TraD, coupling protein.
We thank this reviewer for calling our attention on this work. R1 example and the corresponding citation (Lang et al., 2011) have been added to the paragraph.
- Line 388 Authors write “Mahendra and co-workers …”
In the section 4 of the manuscript the authors use a different way to refer to published work e.g. line 656 “Røder et al. indeed…”. This should be more uniform within the manuscript.
All reference to published works included in the main text are now indicated in the form of “Røder et al.”
- Line 403 Authors write “…plasmid ssb (ssbP) and psiB genes are expressed early upon entry of the plasmid in the transconjugant bacteria…”The DNA enter recipient cell/bacteria and not transconjugant.
This sentence has been corrected accordingly. Thanks for pointing out this lack of accuracy.
- Line 404-405 Authors write, “This was demonstrated using lacZ fusions of the two genes inside the ColIb-P9 plasmid, and also by immunoassays concerning the F plasmid PsiB protein.” Stating that lacZ fusions of two genes was used to demonstrate expression is not enough. This should be either clearly explained or not mentioned at all.
This sentence has been removed as it did not provide any essential information.
- Line 446 Authors write, “SSBC is an essential conserved protein…” Essential for what? Conserved where?
SSB is the single-stranded protein, conserved in all organisms. It is essential to the cell viability as it protects the ssDNA associated with the replication forks. The text now reads: “SSBC is a universally conserved essential protein that binds non-specifically to ssDNA
- Line 448 Authors write, “SSB binding protects the ssDNA…” In the view of the reviewer this is another example of the confusing style used by the authors. This sentence would be much clearer if they would write for example “Upon binding SSB protects the ssDNA…” It sound like a small change but makes a huge difference.
This sentence has been corrected accordingly.
Another example. Line 458 authors write “SSBP can bind ssDNA without specificity…” What about “SSBp binds ssDNA nonspecifically…”
This sentence has been corrected accordingly.
Another sentence with few issues. Line 460 -461 Authors write “However, although producing plasmid F SSBP in trans enables the growth of ssbC deletion mutant, complemented mutant exhibit some filamentation, growth rate reduction.”
- It is either “However” or “although”
This has been corrected.
- “producing plasmid F SSBP” instead “expression of F plasmid encoded SSBP protein”
This has been corrected.
- “…growth rate reduction” ???? do the authors meant “…at reduced growth rate” ???
This has been corrected.
- Line 488-491 Authors write “Here we would like to stress the role of replication in conjugation host-range specificity. Indeed, the ability of the transferred plasmid to replicate in the recipient cells is considered as the primary determinant of plasmid host-range specificity and efficiency of conjugation.” The two sentences here are basically stating the same thing!
The sentence “Indeed (…) conjugation has been removed.
- Line 508 Authors write ”… timely segregation…”what does it mean that segregation is timely???
The term “timely” has been removed
- Line 527 Authors write “…phenotypic conversion of the transconjugants into a novel type of cell that exhibits additional metabolic…” However in the Figure 1 under the Box “Phenotypic conversion” they show conversion of the recipient into a transconjugant??? It will leave the reader very confused.
The text now reads “The expression of genes carried by the newly acquired genetic element results in the phenotypic conversion of the recipient cell into a transconjugant that exhibits additional metabolic properties”
- Line 531 Authors write, “Other plasmid genes encoded by the tra region of the plasmid backbone are not directly involved in the transfer process per se.” The authors need to specify what they mean by other plasmid encoded genes?? Do they want to say “not all the genes encoded within tra region are directly involved in the process of DNA transfer” ??? Very confusing
Indeed, this reviewer is right. This sentence has been corrected accordingly.
- Line 534 authors write, “These immunity systems limit the plasmid-carrying cell ability to serve…”
- Line 541 Authors write, “TraT production reduces the recipient…” Reduces?? What does this mean?
The text now reads “TraT production inhibits the formation (…)”
- Line 551 Authors write “…plasmid transfer ability immunity…” To many nouns!
This has been corrected to “(…) transconjugant cells concomitantly acquire plasmid transfer ability and immunity to self-transfer during the phenotypic conversion”
- Line 594 Authors write “…researches studied…”
This has been corrected.
- Line 661 Authors write “Dynamics of conjugation within biofilms need to be more investigated…” This sounds more like a conclusion rather than an opening statement.
This opening statement has been rephrased “Interactions between conjugation and biofilms have been proposed to promote both community-building and gene transfer. This synergic interaction raises serious questions about the contribution of HGT to the evolution and adaptation of biofilm-forming pathogens”
- Line 665 Authors write “…recent advances tried to bring…” Another awkward phrasing
The text now reads: « recent investigations have aimed to provide a new understanding of biofilm responses to antimicrobial treatments”
- Line 691-692 Authors write “In parallel, antibiotics have also been recognized as signaling molecules increasing conjugative transfer although not all antibiotics can potentiate the frequency of conjugation.” The authors could write “…some antibiotics…” and then the whole part of the sentence coming after conjugative transfer is redundant.
This has been corrected accordingly.
- Line 706-708 Authors write “Lopatkin et al. also demonstrated that antibiotics, from six major classes, not significantly increase the conjugation efficiency of plasmids from five different incompatibility groups.” “Not significantly increase” means that they have no effect!
This has been corrected accordingly.
What is “plasmid-carrying cell ability” ???
This has been rephrased “ (…) limit the ability of plasmid-carrying cells to serve as a recipient for the same plasmid.
- Line 715 Authors write “ …strengthened the difficulty…” Once again awkward phrasing.
This has been rephrased.
In the Figure 2 the authors draw the arrows of various genes but not up to scale. For a person who is not familiar with the gene organization this might suggest that the gene encoding the regulatory proteins are much larger than the ones encoding the actual T4SS components. This needs to be changed.
We provide a revised version of Figure 2 where the scale of the first and last gene of the operon are at scale. The other genes are represented by boxes, since they cannot be show at the tru scale (the operon is very large). The figure legend now explains this representation.
There are more issues related to the style or wording but the reviewer feels that this is too much to ask from him to point all of it out. Therefore, once again it is advised to re-write the manuscript and make it more legible, because in this form it is really very difficult to focus on scientific aspect of this manuscript.
As stated above, the manuscript has been extensively edited by MDPI professional English editing service. Once again, we thank this reviewer for assessing our manuscript in-depth despite the numerous spelling errors and awkward phrasings.

Round 2
Reviewer 1 Report
The revised version of the MS is considerably improved and more consistent in terms of the writing. The English is also greatly improved.
I might suggest removing the phrase "also referred to as bacterial sex" from the abstract. Conjugation has very little resemblance to sexual reproduction processes in eukaryotes, in that no new organisms are produced, and that recombination is a rare accident, not a built in strategy of conjugation. I recognize that the literature on conjugation is replete with images borrowed from sex in eukaryotes (sex pili, fertility factor, matings, male E. coli strains and so on), but this type of terminology should really be avoided, especially in an abstract.
Reviewer 2 Report
The reviewer appreciates all the modifications that the authors have made to the manuscript. In this form the reviewer can recommend the manuscript to be accepted. There is only one suggestion that the reviewer forgot to mention during the first round that the authors might consider. There is evidence in the type IV secretion literature that the partitioning system affects the secretion through the type IV secretion machinery. This have been shown in conjugative systems as well as in Agrobacterium and Neisseria:
Hamilton et al. (2005). Neisseria gonorrhoeae secretes chromosomal DNA via a novel type IV secretion system. Mol. Microbiol. 55, 1704–1721. doi: 10.1111/j.1365-2958.2005.04521.x
Atmakuri, K., E. Cascales, et al. (2007). Agrobacterium ParA/MinD-like VirC1 spatially coordinates early conjugative DNA transfer reactions. EMBO J 26(10): 2540-51.
Guynet et al. (2011). The stb Operon Balances the Requirements for Vegetative Stability and Conjugative Transfer of Plasmid R388. PLoS Genet 7(5): e1002073. doi:10.1371/journal.pgen.1002073
Gruber et al. (2016). Conjugative DNA Transfer Is Enhanced by Plasmid R1 Partitioning Proteins Front. Mol. Biosci
The authors mention the partitioning system but only in context of maintaining the DNA in the recipients. The reviewer believe that it might be interesting to mention also the influence of the partitioning system on the bacterial conjugation. However, this is only a suggestion.
Author Response
Thanks for your recommendation to accept our manuscript for publication in Genes. We have considered your suggestion to add a discussion about the links between partition systems and secretion through the T4SS in different systems. In the case of plasmid transfer, it is an interesting point that underlies the balance between the horizontal and the vertical transfer of plasmids. However, we came to the conclusion that these works do not allow the emergence of a consensual interpretation. Some of the indicated references put forward that the deletion of the plasmid partition systems increases conjugation by modifying the plasmid intracellular position, while others seem to be related to protein-protein interactions. In this context, we have not decided to mention this interplay in our review.